# Claudin-12 Knockout Mice Demonstrate Reduced Proximal Tubule Calcium Permeability

**DOI:** 10.3390/ijms21062074

**Published:** 2020-03-18

**Authors:** Allein Plain, Wanling Pan, Deborah O’Neill, Megan Ure, Megan R. Beggs, Maikel Farhan, Henrik Dimke, Emmanuelle Cordat, R. Todd Alexander

**Affiliations:** 1Department of Physiology, The University of Alberta, Edmonton, AB T6J 2R7, Canada; aplain@gmail.com (A.P.); pan2@ualberta.ca (W.P.); oneilld@ualberta.ca (D.O.); ure@ualberta.ca (M.U.); mbeggs@ualberta.ca (M.R.B.); cordat@ualberta.ca (E.C.); 2The Women’s & Children’s Health Research Institute, 11405-87 Avenue, Edmonton, AB T6G 1C9 Canada; maikel@ualberta.ca; 3Department of Pediatrics, The University of Alberta, Edmonton, AB T6J 2R7, Canada; 4Department of Cardiovascular and Renal Research, Institute of Molecular Medicine, University of Southern Denmark, 5230 Odense, Denmark; hdimke@health.sdu.dk; 5Department of Nephrology, Odense University Hospital, 5000 Odense, Denmark

**Keywords:** proximal tubule, calcium permeability, claudin-12

## Abstract

The renal proximal tubule (PT) is responsible for the reabsorption of approximately 65% of filtered calcium, primarily via a paracellular pathway. However, which protein(s) contribute this paracellular calcium pore is not known. The claudin family of tight junction proteins confers permeability properties to an epithelium. Claudin-12 is expressed in the kidney and when overexpressed in cell culture contributes paracellular calcium permeability (P_Ca_). We therefore examined claudin-12 renal localization and its contribution to tubular paracellular calcium permeability. Claudin-12 null mice (KO) were generated by replacing the single coding exon with β-galactosidase from *Escherichia coli*. X-gal staining revealed that claudin-12 promoter activity colocalized with aquaporin-1, consistent with the expression in the PT. PTs were microperfused ex vivo and P_Ca_ was measured. P_Ca_ in PTs from KO mice was significantly reduced compared with WT mice. However, urinary calcium excretion was not different between genotypes, including those on different calcium containing diets. To assess downstream compensation, we examined renal mRNA expression. Claudin-14 expression, a blocker of P_Ca_ in the thick ascending limb (TAL), was reduced in the kidney of KO animals. Thus, claudin-12 is expressed in the PT, where it confers paracellular calcium permeability. In the absence of claudin-12, reduced claudin-14 expression in the TAL may compensate for reduced PT calcium reabsorption.

## 1. Introduction

Calcium is essential for a myriad of physiological functions including intracellular signal transduction, blood clotting, and as a structural component of bone. It is therefore tightly maintained within a narrow physiologic range in serum. This is achieved through hormonal regulation by parathyroid hormone (PTH), vitamin D, and fibroblast growth factor 23 (FGF23). These hormones, in turn, mediate coordinated interactions between intestinal calcium absorption/secretion, bone resorption/deposition and filtration at the renal glomerulus, and consequent reabsorption along the nephron. The majority, approximately two-thirds of filtered calcium, is reabsorbed by the proximal tubule [1]. This occurs via a paracellular route, primarily driven by the reabsorption of water from the proximal tubule [2,3,4,5,6]. A failure to reabsorb calcium from the proximal tubule has been implicated in the pathogenesis of kidney stone formation [7]. Thus, understanding the molecular mediators of this process is a prerequisite to finding improved therapies for this disease.

Paracellular fluxes not only rely on a driving force, but are also dependent on the permeability of the epithelium to the ion being transported. The proximal tubule has significant permeability to Ca^2+^, enabling paracellular Ca^2+^ flux [1,3,4,5]. Claudins are a family of four pass membrane proteins expressed in the tight junction that confer paracellular permeability to the tight junction [8]. Claudin-2 is expressed in the proximal tubule and claudin-2 knockout (KO) mice display increased urinary calcium excretion relative to their wild type littermates, consistent with claudin-2 conferring calcium permeability to the proximal tubule [8,9]. Claudin-2 and claudin-12 are expressed in the intestine, and are implicated in mediating calcium permeability [10,11]. Claudin-12 mRNA has been detected in the kidney [2,12]. We have previously reported claudin-12 expression in a renal proximal tubular cell culture model, opossum kidney (OK) cells [13], and mRNA expression has been identified in proximal tubules, where it could also contribute paracellular Ca^2+^ permeability [14].

To study claudin-12 renal expression and if it contributes Ca^2+^ permeability to the renal tubule, claudin-12 KO mice were generated by replacing the single coding exon with β-galactosidase from *E. coli*. The knockout mice grew and behaved similarly to their wild type (WT) littermates. We did not detect differences in plasma electrolytes nor in calciotropic hormone levels between KO and WT mice. We observed predominant X-gal staining in the renal cortex that colocalized with aquaporin-1, indicating expression of claudin-12 in the renal proximal tubule. We therefore perfused proximal tubules from claudin-12 KO mice and found reduced paracellular Ca^2+^ permeability. The KO mice did not however have increased urinary Ca^2+^ excretion. We thus examined the expression of genes that participate in tubular Ca^2+^ transport and found decreased expression of the paracellular Ca^2+^ blocker, claudin-14 [15], and propose that increased thick ascending limb (TAL) Ca^2+^ reabsorption compensates for reduced proximal tubular Ca^2+^ reabsorption in claudin-12 knockout mice.

## 2. Results

### 2.1. Generation of a Global Claudin-12 Knockout Model

In order to examine the renal localization and potential tubular transport role of claudin-12, we generated a global claudin-12 knockout mouse. The claudin-12 gene (*Cldn12*) was replaced by homologous recombination of exon 4, the only coding exon of the *Cld12* gene, with the β-galactosidase coding sequence from *E. coli* (Figure 1A). Specific PCR reactions failed to amplify the wild type sequence from KO animals. Similarly, PCR with primers specific for β-galactosidase did not amplify a product from wild type DNA. However, appropriate size PCR fragments could be amplified from both the wild type *Cld12* gene and β-galactosidase from heterozygous mice (Figure 1C). Moreover, quantitative real-time PCR performed on cDNA generated from RNA isolated from whole kidney of wild type mice detected *Cld12* that was not detectable in knockout mice (Figure 1B), consistent with known renal *Cld12* expression [2]. Unfortunately, we have been unable to identify a commercially available antibody or generate an antibody that was specific for claudin-12, as was previously reported by Professor Tsukita’s group [16], and more recently by a group studying claudin-12 in nervous tissue [17]. We therefore performed a sequencing reaction on DNA from wild type and claudin-12 KO mice using a primer specific for a sequence approximately 50 bp 5′ to the start cogon in exon 4 of claudin-12. This confirmed that the claudin-12 coding sequence had been replaced with β-galactosidase (Figure 1D).

### 2.2. Claudin-12 is Expressed in the Renal Proximal Tubule

In order to localize claudin-12 expression in the kidney, we performed X-gal staining of kidney sections on wild type mice and heterozygous littermates (Figure 2). No staining was evident on the wild type kidney section (Figure 2A). However, we observed prominent staining in the knockout renal cortex, consistent with claudin-12 promoter activity (Figure 2B). Interestingly, the greatest amount of color production was in the juxtamedullary region, with virtually no color produced in the medulla. Higher power images of the cortical region clearly demonstrate β-galactosidase expression concentrated in some, but not all, of the tubules present in the juxtamedullary cortex (Figure 2C). The tubules with greatest β-galactosidase expression were the largest, had an obvious brush border, and were the most abundant in the cortex, consistent with claudin-12 promoter activity in the proximal tubule.

In order to more precisely define claudin-12 renal cortical expression, we performed immunofluorescence localization with tubule segment-specific markers on renal sections from heterozygous mice, after X-gal staining. Aquaporin I (AQP1), a marker of the proximal tubule in the renal cortex [18], colocalized with X-gal production (Figure 3A), confirming predominant localization to the proximal tubule. AQP1 is also present in the thin descending limb in loop of Henle [18], but we limited our assessment to the cortex. For the identification of the thick ascending limb, distal convoluted tubule, and collecting duct, we performed immunostaining with the sodium–potassium–chloride cotransporter II (NKCC2, Figure 3B), the sodium–chloride cotransporter (NCC, Figure 3C), and carbonic anhydrase II (CAII, Figure 3D), respectively. Although CAII is present in the proximal tubule, expression is very low and the greatest expression by far is in the collecting duct (intercalated cells); thus, CAII can be used as a marker of this segment [19]. We observed AQP1 co-staining with X-gal stained tubules. However, none of the other markers demonstrated significant X-gal co-staining, which is consistent with predominant, if not exclusive, expression of claudin-12 in the renal proximal tubule. Importantly, this method of colocalization cannot exclude low levels of claudin-12 mRNA expression.

### 2.3. Claudin-12 Deletion Decreases Sodium and Calcium Permeability of the Proximal Tubule

We next turned our attention to the putative role of claudin-12 in the proximal tubule. To this end, we microperfused freshly isolated straight proximal tubules (Figure 4A) from either wild type (WT) or knockout (KO) mice, as this is the segment which demonstrated the most intense X-gal staining. We recorded transepithelial voltage across the tubule (Figure 4B,C) and used a current pulse (13 nA) to calculate the transepithelial resistance. The transepithelial resistance, transepithelial voltage, and short-circuit current determined before and after the addition of ouabain were indistinguishable between WT and KO animals (Figure 4D–F). Ouabain inhibits the Na^+^/K^+^ATPase, eliminating transepithelial Na^+^ transport (as it is the driving force for vectorial Na^+^ flux). Importantly, as we were applying a dilution potential across the tubule to measure permeability, not flux, this did not alter our measurement. Ouabain dramatically reduced the transepithelial voltage and consequently the short-circuit current. This is consistent with transcellular transport being substantially reduced, as would be expected. Thus, any subsequent changes in transepithelial voltage generated after the addition of ouabain are primarily due to the paracellular movement of ions induced by the change in the basolateral solution composition. Importantly, the decrease in transcellular transport was not statistically different between WT and KO tubules.

Next, we imposed a sodium chloride concentration gradient, by adding a solution containing low sodium to the basolateral side, and measured the diffusion potential generated across the tubule. With this result and the resistance, we calculated the permeability ratio of sodium to chloride (Figure 5A, P_Na_/P_Cl_). Interestingly we observed an increase in the potential difference (PD) across the tubule in the KO mice, as opposed to a decrease in PD in the WT animals, consistent with altered relative sodium to chloride permeability. We then imposed a calcium to sodium diffusion gradient across the tubule, recorded the diffusion potential generated and used this to calculate the permeability ratio of calcium to sodium (Figure 5A, P_Ca_/P_Na_). From these ratios and the resistance, we were able to calculate the absolute permeabilities of the tubule to sodium, chloride, and calcium (Figure 5B). Straight proximal tubules from claudin-12 KO mice displayed a reduced sodium relative to chloride permeability ratio (1.27 ± 0,05 in the WT compared with 0.89 ± 0.17 in the KO), which was the result of decreased sodium permeability (3.8 ± 1.3 × 10^−4^ cm/s in WT vs. 2.5 ± 0.4 × 10^−4^ cm/s in KO), and not increased chloride permeability (Figure 5). Hence, the deletion of claudin-12 changes the selectivity of the straight proximal tubule to relatively more anion permeable, i.e., claudin-12 confers cation selectivity to the straight portion of the renal proximal tubule.

We next examined the calcium permeability characteristics of the proximal tubule. Interestingly, straight proximal tubules from wild type mice displayed a calcium to sodium permeability ratio of about 2 (P_Ca_/P_Na_ = 1.87 ± 0.04, Figure 5A), inferring that this part of the nephron is approximately twice as permeable to calcium as to sodium. Further, the deletion of claudin-12 resulted in a greater reduction in permeability to calcium than to sodium (P_Ca_/P_Na_ = 1.51 ± 0.08, Figure 5A), which was also reflected in decreased absolute calcium permeability (7.1 ± 0.8 × 10^−4^ cm/s in the WT compared with 3.8 ± 0.5 × 10^−4^ cm/s in the KO) (Figure 5B, P_Ca_). These results indicate that the straight portion of the proximal tubule has selective calcium permeability and that claudin-12 confers some, but not all, of the selective and total permeability to calcium.

### 2.4. Claudin-12 Null Mice Do not Have Hypercalciuria

We hypothesized that reduced calcium permeability in the proximal tubule of claudin-12 knockout mice would result in increased urinary calcium excretion and/or hormonal compensation. We therefore housed mice in metabolic cages with water and chow provided ad libitum. The claudin-12 knockout mice weighed the same as their wild type littermates and no differences were observed in water or chow ingested, urine volume, or fecal weight (Table 1). Further, analysis of serum at the end of the metabolic cage experiments failed to identify significant differences in plasma electrolytes, including Ca^2+^, creatinine, glucose, and blood urea nitrogen (BUN) (Table 2). Importantly, we did not find differences in the circulating levels of parathyroid hormone (PTH), FGF23, nor the active form of vitamin D, 1,25 dihydroxy vitamin D_3_/calcitriol (Table 2). Finally, we examined urinary and fecal excretion. We did not detect a significant difference in the urinary excretion of all electrolytes examined, including Ca^2+^ and Na^+^ (Table 3). We also did not observe an alteration in urinary excretion of creatinine, consistent with the same glomerular filtration rate in both genotypes (Table 3). In summary, claudin-12 knockout mice, despite reduced proximal tubular Na^+^ and Ca^2+^ permeability, do not display altered urinary excretion of these ions, or hormonal compensation.

Wild type and knockout mice were then fed a normal (0.6% wt/wt), low (0.01% wt/wt) or high (2.0% wt/wt) calcium diet, to evaluate if altering calcium intake would alter urinary calcium excretion (Table 4). The diets increased or reduced urinary calcium excretion as expected, but no clear differences between genotypes were observed for either FE_Ca_ or plasma Ca^2+^ levels between genotypes on the different diets.

### 2.5. Renal Compensation in Claudin-12 KO Mice

Having failed to identify urinary wasting of Ca^2+^ or Na^+^ or apparent hormonal compensation, we turned our attention to possible intrarenal compensatory mechanisms. Firstly, we examined the cortical protein expression of Claudin-10 and -2, which are also known to be expressed in the proximal tubule (Figure 6A–D). We found that both were significantly reduced in expression, in the KO mice.

We also isolated RNA from the whole kidney of wild type and claudin-12 KO mice and performed a quantitative real-time PCR. We did not find a significant difference in the expression of genes mediating transcellular Ca^2+^ reabsorption from the distal convoluted tubule except for PMCA1b, which was slightly reduced (Figure 7A). We also failed to identify significant differences in the expression of apical sodium transport proteins along the nephron (Figure 7B). Further, consistent with the absence of a significant difference in circulating calcitriol levels, we did not find differences in the expression of renal enzymes regulating the amount of this hormone in plasma, the vitamin D receptor, the calcium sensing receptor (CaSR), nor in klotho (Figure 7C). Finally, we assessed the expression of renal claudins (Figure 7D) and found that claudin-1, -8, and -14 had a significantly reduced expression in the knockout animals. Claudin-14 reduction is particularly interesting as it is only expressed in the cortical TAL [20], the nephron segment responsible for the reabsorption of around 25% of filtered calcium [1]. When present, claudin-14 attenuates calcium reabsorption from the TAL via the paracellular pathway [15]. Reduced expression in the knockout mice may therefore reflect increased TAL calcium reabsorption, in compensation for decreased proximal tubule calcium reabsorption.

## 3. Discussion

Evidence supports a role for claudin-12 in contributing paracellular Ca^2+^ permeability in the intestine [11]. Claudin-12 mRNA is expressed in the kidney, and a proximal tubule cell culture model [10,13,21]. The proximal tubule is the location of the greatest amount of Ca^2+^ reabsorption along the nephron and this occurs via the paracellular pathway [1,3,4,5]. To date the tight junction proteins contributing calcium permeability to this segment have been incompletely defined. We therefore hypothesized that claudin-12 contributes paracellular permeability to the proximal tubule and generated a claudin-12 KO mouse, by replacing the only coding exon of the claudin-12 gene with β-galactosidase to assess this possibility. This model enabled us to localize *Cldn12* to the renal proximal tubule, and by perfusing tubules ex vivo from wild type mice and claudin-12 KO animals, we confirmed that *Cldn12* does indeed contribute Ca^2+^ and Na^+^ permeability to this segment. However, the deletion of *Cldn12* does not induce hypercalciuria or an increase in calciotropic hormone levels.

Direct measurement of Ca^2+^ permeability of the straight portion of the proximal tubule of wild type and claudin-12 KO mice revealed relative selectivity for Ca^2+^ over Na^+^, which was attenuated in the absence of claudin-12. This strongly supports *Cldn12* contributing to the Ca^2+^ permeability of the proximal tubule. It is surprising then that claudin-12 null mice did not display hypercalciuria in response to a failure to reabsorb Ca^2+^ from the proximal straight tubule. Further, there appeared to be no systemic compensation for reduced paracellular Ca^2+^ permeability in the proximal nephron, as calciotropic hormone levels were also unchanged. What then may explain our observations? Potentially altered permeability of the proximal tubule is compensated by an increased driving force for Ca^2+^ across this segment. This could occur via generating a more lumen positive potential difference across the segment, or increasing water reabsorption from the proximal tubule, which would in turn either enhance the Ca^2+^ concentration gradient or convective driving force for paracellular Ca^2+^ [22]. We did not observe a significant difference in baseline transepithelial voltage (Vte) across tubules perfused ex vivo between wild type and knockout mice (Vte WT = −3.2 ± 0.4 mV, *n* = 9 and KO = −2.5 ± 0.5 mV, *n* = 8, *p* = 0.253), making the former explanation unlikely. Further, we did not observe altered expression of the major Na^+^ transporter in the proximal tubule, sodium hydrogen exchanger isoform 3(NHE3), suggesting that this is not the explanation, although altered NHE3 activity can occur in the absence of altered expression [23,24,25]. Instead, we observed decreased *Cldn14* expression, consistent with increased Ca^2+^ reabsorption from the TAL in compensation. Claudin-14 interacts in the cortical TAL (cTAL) tight junction with the claudin-16/19 complex, reducing permeability to calcium, and thus calcium absorption from this segment [15,26]. This suggests that increased Ca^2+^ reabsorption from the TAL is compensating for reduced proximal tubule Ca^2+^ reabsorption.

How might *Cldn14* expression be altered in the absence of changes in circulating Ca^2+^ or PTH levels? The expression of Cldn14 is increased by the activation of the calcium sensing receptor (CaSR) in the basolateral membrane of the TAL [15]. Plasma Ca^2+^ levels were not different between wild type and claudin-12 KO mice, however, we were unable to assess the local concentration of Ca^2+^ in the peritubular space of the cortical TAL (where *Cldn14* is expressed [20]). Interestingly, the cTAL runs linearly into the juxtamedullary cortex, and Figure 3 reveals that the cTAL directly abuts tubules with significant *Cldn12* promoter activity. Consistent with this observation, microdissection of straight segments of proximal tubules frequently demonstrated the close adherence of a cTAL (Figure 8). Perhaps, then, reduced reabsorption of Ca^2+^ from the straight portion of the proximal tubule, leads to a lower concentration of Ca^2+^ in the juxtamedullary interstitium, and thus reduced CaSR activation? This would result in increased cTAL Ca^2+^ permeability and increased reabsorption from this segment, in compensation. The proposed proximal tubule–cTAL crosstalk would explain our lack of alteration in urinary Ca^2+^ excretion and the lack of hormonal compensation. However, this hypothesis requires further experimental testing to support it.

Claudin-2 has also been implicated in paracellular intestinal Ca^2+^ absorption. In addition, this protein has been clearly localized to the proximal tubule, where it contributes paracellular Na^+^ permeability [9,11,27]. Proximal tubule paracellular Na^+^ permeability contributes a significant amount of net Na^+^ reabsorption from this segment. Using the paracellular pathway and the concentration gradient/convection forces induced by the considerable water flux across this segment increases the efficiency of this process, thereby reducing the energy required [28]. Interestingly, claudin-2 KO mice display hypercalciuria [9]. It is therefore tempting to speculate that claudin-2 also contributes proximal tubule Ca^2+^ permeability as we observe for claudin-12. Further specific studies will be required to support this. Why claudin-2 KO mice demonstrate hypercalciuria and claudin-12 null mice do not is unclear, but is perhaps due to a greater abundance of claudin-2 in the proximal tubule.

PTH is released from the parathyroid gland in response to low plasma calcium levels. It acts directly on the proximal tubule to inhibit transcellular sodium reabsorption and likely paracellular Ca2^+^ reabsorption [22]. It also acts on the distal convoluted tubule/connecting tubule to increase Ca^2+^ reabsorption [29]. FGF23 is another phosphocalciotropic hormone that acts on the proximal tubule to inhibit sodium phosphate cotransport, and on the distal convoluted tubule/connecting tubule to augment Ca^2+^ absorption [30,31]. Although we did not detect a significant difference in the circulating amounts of these hormones between wild type and claudin-12 KO mice, there were large variations in plasma levels. It is thus possible that alterations in these hormones could be compensating for the reduced paracellular Ca^2+^ permeability in the proximal tubule.

The lack of a specific claudin-12 antibody has limited our ability to confirm protein localization. This is despite trying several commercially available antibodies and generating two of our own. It should be emphasized that we are not the only group to experience this difficulty [12,17]. Regardless, we have shown clearly the absence of claudin-12 DNA in the knockout mice and the inability to detect messenger RNA in the kidney of knockout mice. Moreover, the claudin-12 KO mice that we generated do not have alterations in the endogenous promoter, and we replaced the coding sequence with a reporter β-galactosidase, which clearly demonstrates an expression in the renal cortex, as seen in Figure 2. We are therefore confident that *Cldn12* is expressed in the renal proximal tubule where it confers Ca^2+^ permeability. We should emphasize that we cannot exclude lower levels of expression in other nephron segments.

In conclusion, using a mutant mouse engineered to have β-galactosidase replace the coding exon of *Cldn12*, we were able to localize claudin-12 expression to the renal proximal tubule. Microperfusion studies on this model revealed reduced Na^+^ and Ca^2+^ permeability consistent with claudin-12 conferring paracellular Ca^2+^ permeability to the proximal tubule. Surprisingly the knockout mice did not have increased urinary Ca^2+^ excretion, or alterations in calciotropic hormone levels. We propose that this is due to a proximal tubule–TAL crosstalk leading to reduced *Cldn14* expression and consequently increased Ca^2+^ reabsorption from the TAL, in compensation for decreased proximal tubule Ca^2+^ reabsorption.

## 4. Materials and Methods

### 4.1. Generation of Cld12 KO Mice

A claudin-12 null strain was generated through the UC Davies Knock Out Mouse Project (KOMP). The gene encoding claudin-12 is located on chromosome 5, and contains four exons with the coding sequence encoded by exon four (NCBI Gene: 64945). A targeting vector (Velocigene cassette ZEN-Ub1, KOMP Repository category number 13208L1) was used to replace exon four with the neomycin resistance gene and the lacZ reporter gene by homologous recombination in VGB6 embryonic stem (ES) cells derived from the mouse strain C57BL/6NTac. The mouse strain used for this research project, KOMP ES cell line Cldn12^tm1(KOMP)Vlcg^, RRID:MMRRC_053773-UCD, was obtained from the Mutant Mouse Resource and Research Center (MMRRC) at the University of California at Davis, an NIH-funded strain repository, and was donated to the MMRRC by The KOMP Repository, UC Davis Mouse Biology Program. This created a 760-bp deletion between positions 5507663–5508422 of Chromosome 5 resulting in heterozygotes (HET) ES cells (Figure 1A). Subsequently, the recombinant ES cells were used to generate HET mice. Littermates were used as controls. Intercrossing of heterozygotes yielded claudin-12 deficient mice (KO). Genotypes were confirmed by RT-PCR, and gene sequencing (Figure 1B–D).

### 4.2. Metabolic Cage Studies

Metabolic Cage studies were performed as previously reported [2,15]. Three month-old wild type (WT), or claudin-12 deficient (KO) mice were placed in metabolic cages for three days with free access to normal rodent diet chow (LabDiet^®^ 5001, Fort Worth, TX, USA) and water. Weight, chow eaten, and water consumed were measured daily. Urine and feces were collected every 24 h for analysis. Only samples from day three were used for analysis. Then, animals were anesthetized using pentobarbital sodium, and blood was collected for analysis of serum creatinine via high-performance liquid chromatography. We also performed a blood gas at that time. Kidneys were harvested and snap frozen in liquid nitrogen and stored at −80 °C. Metabolic cage studies on wild type and claudin-12 KO mice fed different calcium containing diets (low = 0.01% TD. 95027, normal = 0.6% TD.97191, or high = 2% TD.00374, all from Harlan Laboratories, Madison, WI, USA) were performed as previously described [15]. All experimental procedures were approved on 3 October 2013 by the Animal Care and Use Committee for Health Sciences at the University of Alberta (protocol number 00000213).

### 4.3. Measurement of Urinary, Fecal, and Plasma Electrolytes

Serum electrolytes, blood urea nitrogen, and glucose were measured with a handheld blood gas analyzer (Vet Scan i-STAT1 Analyzer, Abaxis, Union City, CA, USA) as previously described [15]. Feces from the last 24-h collection were dried for 48 h in an incubator (Labline, Imperial III Incubator, Mumbai, India) at 55 °C, and then homogenized with a mortar and pestle. A sample of 0.7 g per mouse was then taken and solubilized in nitric acid for elemental analysis. Nitric acid (70%) was added to the samples in two sequential steps: the first 0.3 mL were added when the sample was at 65 °C, then 0.7 mL were added once fumes subsided after about 30 min. The mix was then heated to 85 °C and boiled until the fumes stopped (approximately 1 h later). Subsequently, 1 mL of 30% H_2_O_2_ was added, again in two steps: first, 0.2 mL were added while the sample was still at 85 °C. Once boiling stopped, the sample was allowed to cool for 10 min before adding the remaining 0.8 mL. After another 10 min at room temperature, the samples were heated again to 85 °C until the reaction stopped (approximately 1 h later). The samples were adjusted to 5 mL with ddH_2_O and used for analysis. The calcium content was measured using the QuantiChrom Ca^2+^ Assay Kit (Catalog number: DICA-500) from BioAssay Systems (Hayward, CA, USA).

Urine electrolyte content was measured by ion chromatography. All experiments were performed with a Dionex Aquion Ion Chromatography (IC) System (category number 22176-60004, Thermo Fisher Scientific Inc., Mississauga, ON, Canada) equipped with an autosampler. The cation and anion eluents were prepared as per the manufacturer’s instructions. The anion eluent consisted of a 4.5 mM Na_2_CO_3_/1.5 mM NaHCO_3_ in ddH_2_O, and the cation eluent consisted of 20 mM Methanesulfonic acid in ddH2O. All reagents and samples were filtered with a 0.2 µm syringe filter (category number 03-391-1B, Thermo Fisher Scientific Inc., Mississauga, ON, Canada), and stored in high density polyethylene containers that had been thoroughly cleaned with deionized water to avoid any traces of ions. Calibration standard curves were produced using Dionex five anion standard (category number 037157, Thermo Fisher Scientific Inc., Mississauga, ON, Canada) and Dionex six cation-I standard (category number 040187, Thermo Fisher Scientific Inc., Mississauga, ON, Canada). Thermo Scientific Chromeleon 7 Chromatography Data System (CDS) software was used for automation and data handling. High-performance liquid chromatography (HPLC) was used to detect creatinine concentration in urine and blood samples. All experiments were performed with a Dionex UltiMate 3000 HPLC System (Thermo Scientific, ISO-3100SD PUMP category number 5040.0011; TCC-3000SD Column Thermostat, category number 5730.0010, and VWD-3100 DETECTOR, category number 5074.0005, Thermo Fisher Scientific Inc., Mississauga, ON, Canada). UV detection of samples occurred at 216 nm and the pump flow rate was set at 0.2 mL/min accordingly. The HPLC eluent was prepared as per the manufacturer’s instructions. The eluent consisted of acetonitrile (category number A-0626-17, Thermo Scientific)/Trifluoroacetic acid category number T-3258-PB05, Thermo Scientific, Mississauga, ON, Canada) in ddH_2_O. All reagents and samples were filtered using 0.2 µM syringe filters (Thermo Scientific, category number 03-391-1B, Mississauga, ON, Canada) and stored in glass containers that had been thoroughly cleaned with deionized water to avoid sample adsorption. Calibration standard curves were produced using creatinine from Acros Organics, NJ, USA, category number AC228940250. Thermo Scientific Chromeleon 7 Chromatography Data System (CDS) software was used for automation and data handling.

### 4.4. Measurement of Hormone Levels

PTH was measured by ELISA according the manufacturer’s instructions (mouse PTH 1-84 ELISA Kit, Immutopics Inc., category number 60-2305, San Diego, CA, USA) and Vitamin D levels were determined by a radioimmunoassay (1,25-Dihydroxy Vitamin D RIA kit, Immunodiagnostic systems, category number AA-54F1, Immunodiagnostic Systems Inc, Gaithersburg, MD, USA), according to the manufacturer’s instructions as previously reported [2,15]. FGF23 was quantified by ELISA according to the manufacturer’s instructions (FGF-23 ELISA Kit, Kainos, category number CY-4000, Tokyo, Japan).

### 4.5. X-Gal Staining and Immunofluorescence Microscopy on Renal Sections

We performed X-gal staining on 8-µm renal cryosections fixed with periodate–lysine–paraformaldehyde and prepared as previously reported [2]. Sections were first washed three times with fresh phosphate buffered saline, then rinsed briefly with distilled water. Next sections were incubated for approximately 3 h in X-gal solution, which consisted of a 1:40 dilution of pure X-gal (Sigma-Aldrich, Oakville ON, Canada) dissolved in 4% DMSO with a dilution buffer consisting of potassium ferricyanide crystalline 5 mM, potassium ferricyanide trihydrate 5 mM, MgCl_2_ 2 mM, and PBS 100 mM, to allow the development of a blue color. Wild type littermate sections were taken as a negative control. Then, sections were rinsed with PBS twice for 5 min before mounting with DAKO (Carpinteria, CA, USA) or undergoing immunostaining, essentially as previously reported [19,32]. In brief, antigen retrieval was performed with sodium citrate, and then washed three times with TN buffer (0.1 M Tris/HCl and 0.15 M NaCl, pH 7.6). Then the section was incubated with 0.3% H_2_O_2_ in TN buffer for 30 min. Sections were then washed three times with TN buffer before blocking for 1 h with TN buffer containing 0.5% Blocking Reagent (Perkin Elmer, MA, USA) and 0.05% Tween 20. Next, primary antibodies were applied in TN buffer (1:1500 anti-NKCC2, Developmental Studies Hybridoma Bank University of Iowa, USA, 1:500 anti- CAII (Santa Cruz Biotechnology Inc, Dallas, TX, USA, 1:500 anti-AQP1, Santa Cruz Biotechnology Inc., Dallas, TX USA or 1:500 Anti-NCC, Stress Marq Biosciences Inc. Victoria, BC, Canada), overnight all diluted in TN buffer. The slides were then washed three times with TN buffer and then a Cy3 conjugated secondary antibody of either donkey anti-mouse (NKCC2, CAII), anti-rabbit (NCC), or anti-goat (AQP1) was applied in the TN buffer containing 0.05% Tween-20 (TNT) for an hour. Finally, the slides were washed three times with TNT buffer before being mounted and visualized with an Olympus BX51 microscope equipped with a X-cite Series 120Q light source (Lumen Dynamics Inc. Mississauga, ON, Canada) and an Infinity 3, 1.4 Megapixel Cooled, color Camera (Lumenera, Ottawa, ON, Canada).

### 4.6. Proximal Tubular Perfusion and Electrophysiology

Proximal tubules were isolated and microperfused essentially as previously described [33]. To this end, kidneys were collected from three month-old claudin-12 WT and KO mice, swiftly decapsulated and sliced transversely. Cortical and juxtamedullary straight proximal tubules where mechanically dissected from the slices at 4 °C on a Leica M165C dissecting microscope and transferred to an Axi Observer Zeiss microscope for microperfusion. A Vestavia Scientific, LLC (Atlanta, GA, USA) microperfusion system of concentric micropipettes was used to hold and microperfuse the tubules as described previously [34]. The glass pipettes where built on a Luigs & Neumann GmbH microforge (Ratingen, Germany). We used for the external holding pipettes sodalime glass capillaries (Hilgenberg, Germany, category number 1409679), and elongated double-barreled glass capillary (Theta-Bo-glasscapillaries, Hilgenberg, Germany, category number 1402401) for the double-barreled perfusion pipettes. A silver wire of 0.2-mm diameter was inserted in one of the barrels of the perfusion pipette to be later connected to a pulse generator. We injected currents of 13 nA for one second every other second via a 150-MΩ resistor into the perfused tubules. The generated voltage deflections where used to later calculate the transepithelial resistance using the cable equation. The perfusion system was connected to a pressure pump that was designed at the Institute of Physiology of the University of Kiel, Germany, to keep a constant luminal flow through the perfused tubules. The bath and tubule lumen (via one of the barrels of the perfusion pipette) where connected via agar bridges (3M KCl, 3% agar) to silver–chloride electrodes, Ag/AgCl (RC-3, World Precision Instruments, Sarasota, FL, USA), which were themselves connected to ground, an amplifier, and to an impedance converter. The signal was digitalized with a PowerLab 8/SP and recorded with LabChart data acquisition and analysis software (LabChart 4.2, ADInstruments–North America, Colorado Springs, CO, USA). This enabled the recording of the transepithelial voltage and the transepithelial voltage deflections throughout the course the experiments. The length and diameter of the tubules were measured on the digital images taken using Zen 2.6 Blue Edition Carl Zeiss Microscopy modular image-processing and analysis software (Carl Zeiss Canada, Toronto, ON, Canada). Images were captured with a Hamamatsu ORCA-Flash4.0 V3 digital camera, model C13440-20CU (Hamamatsu Corporation - Americas, Bridgewater, NJ, USA).

In order to measure calcium permeability, tubules were first perfused with modified Ringer’s control solution (132 mM NaCl, 0.4 mM KH_2_PO_4_, 1.4 mM K_2_HPO_4_, 1.2 mM MgCl_2_ · 6H_2_O, 5 mM Na-Acetate · 3H_2_O, 2.5 mM NaHCO_3_, 1.3 mM CaCl_2_, 5 mM L-alanine, and 5 mM glucose) in a running bath of the same control solution. Subsequently, the bath solution was exchanged to the same control solution containing ouabain (10 µM). Once voltage recordings stabilized, we changed the basolateral solution to first induce a NaCl dilution potential, and then a Na^+^-Ca^2+^ bi-ionic diffusion potential (all the solutions contained 10 µM ouabain) (Figure 4B,C). The 70 mM low sodium solution (65 mM NaCl, 0.4 mM KH_2_PO_4_, 1.4 mM K_2_HPO_4_ · 3H_2_O, 1.2 mM MgCl_2_ · 6H_2_O, 5 mM Na-Acetate · 3H_2_O, 2.5 mM NaHCO_3_, 1.3 mM CaCl_2_, 5 mM L-alanine, 5 mM Glucose, and 112 mannitol) and the 72.5 mM high calcium containing solution (3.6 mM KCl, 72.5 mM CaCl_2_ · 2H_2_O, 5 mM glucose, and 3mM HEPES) were designed using the control solution as the base. The sodium–chloride permeability ratios and the calcium–sodium permeability ratios were then calculated using the Goldmann–Hodgin–Katz flux equation. The absolute permeability to sodium was calculated using the simplified Kimizuka–Koketsu equation, and the absolute permeabilities to chloride and calcium obtained from this value and the permeability ratios [35,36].

### 4.7. Quantitative Real-Time PCR and Immunoblotting

Quantitative real-time PCR and immunoblotting were performed essentially as previously described [2,15,16,21]. Total mRNA was extracted from ½ of a snap frozen kidney from three month-old mice using the TRIzol Reagent (Invitrogen, Carlsbad, CA, USA) according to the manufacturer’s instructions. After treatment with DNAse (ThermoScientific, Vilnius, LT, USA, category number EN052), cDNA was reverse transcribed from the isolated total mRNA. Primers and probes were obtained from Applied Biosystems Inc. (Foster City, CA, USA). A quantity of 5 μL (125 ng of cDNA) was used as a template to determine the gene expression by qPCR. TaqMan universal qPCR master mix (Applied Biosystems Inc, Foster City, CA, USA), primer, probe, and RNAse-free water were mixed together, then added to cDNA that had been placed in a 384-well plate (Applied Biosystems Inc, Foster City, CA, USA). Internal control mRNA levels of 18S ribosomal RNA were measured. Expression levels were quantified with an ABI Prism 7900 HT Sequence Detection System (Applied Biosystems Inc, Foster City, CA, USA), and 18S was used for normalization of RNA as none of the experimental perturbations resulted in a significant change in its expression. Immunoblotting was carried out using an anti-claudin-10 (Invitrogen Inc., category number 38-8400, a subsidy of Thermo Fischer Inc., Mississauga, ON, Canada) or anti-claudin-2 (Thermo Fischer Inc., category number 32-5600, Mississauga, ON, Canada) and appropriate secondary antibodies, and then quantified as previously described [2].

### 4.8. Statistical Analysis

The differences between group mean were assessed by Student’s t test with a Bonferroni correction for multiple comparisons applied when needed. The Shapiro–Wilk test was used to evaluate the data for normal distribution when pertinent. The results are presented as mean ± SEM (*n* = number of animals throughout) and considered significant at *p* < 0.05. Renal mRNA expression was analyzed using the Benjamini–Hochberg procedure, with a critical value for false discovery rate of 0.05 [37].

## Figures and Tables

**Figure 1 ijms-21-02074-f001:**
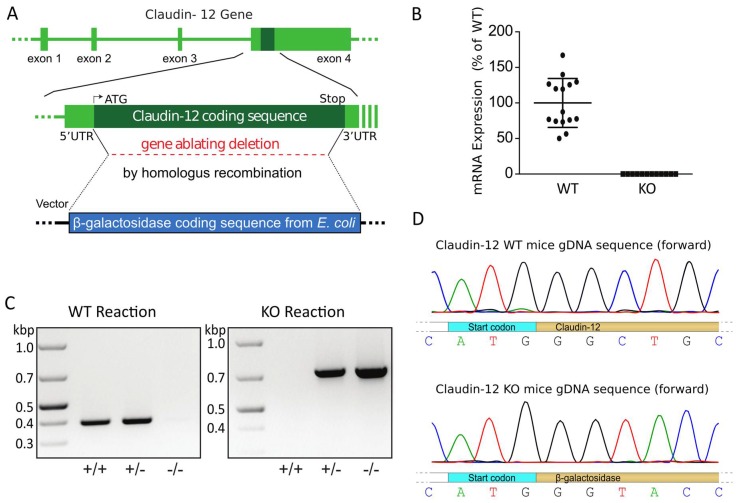
Claudin-12 knockout mouse model. (**A**) Claudin-12 gene deletion targeting strategy. (**B**) Relative claudin-12 mRNA expression normalized to GAPDH, from kidneys of wild type (WT) or claudin-12 knockout mice (KO). (**C**) Genotyping reactions of wild type (+/+), heterozygous (+/−), or claudin-12 knockout (−/−) mice using specific primers for the coding exon of claudin-12 (WT reaction) or the β-galactosidase coding sequence from *E. coli* (KO reaction). (**D**) Sequencing results of the gDNA sequence of claudin-12 WT or KO mice.

**Figure 2 ijms-21-02074-f002:**
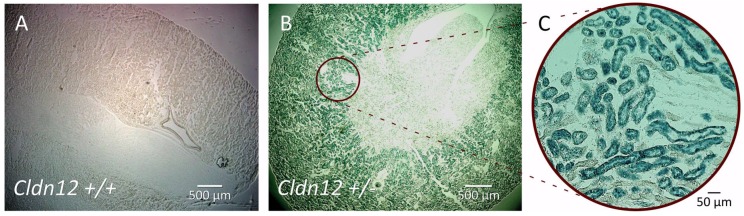
X-gal staining of kidney sections from wild type (**A**) and heterozygous (**B**) mice. β-galactosidase is expressed in the renal cortex in knockout kidney slices and not in wild type ones. Higher power image (**C**).

**Figure 3 ijms-21-02074-f003:**
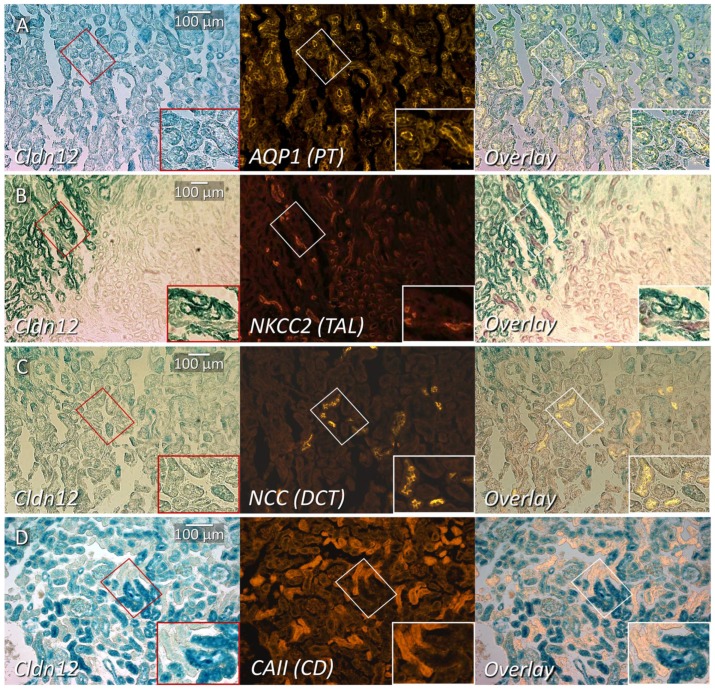
Co-staining of cortical renal tubule markers with X-gal on kidney slices from claudin-12 heterozygous mice. X-gal staining representing claudin-12 promoter expression (Cld12) co-stained with: (**A**) aquaporin-1 (AQP1), (**B**) sodium–potassium–chloride cotransporter 2 (NKCC2), (**C**) sodium–chloride cotransporter (NCC), or (**D**) carbonic anhydrase II (CAII); markers of the proximal tubule (PT), thick ascending limb (TAL), distal collecting tubule (DCT) and collecting duct respectively. The box in the right lower corner is a digitally magnified image from the smaller area depicted with the same shape.

**Figure 4 ijms-21-02074-f004:**
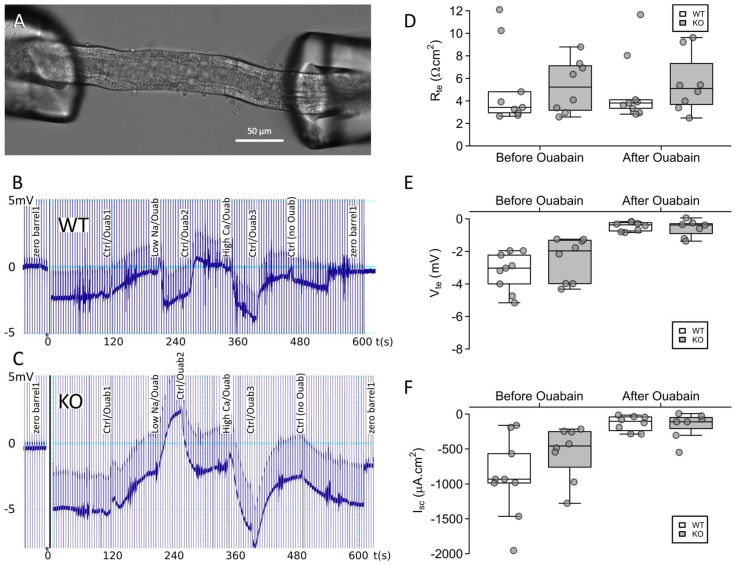
Microperfused proximal tubules from claudin-12 knockout (KO) and wild type (WT) mice. (**A**) Representative image of a proximal tubule perfused ex vivo. (**B**,**C**) Representative recording traces from original experiments in proximal tubules from WT (**B**) and KO (**C**) animals. Basolateral addition of ouabain, and solutions containing a low sodium or high calcium concentration altered the trace by first reducing transcellular transport, and then generating diffusion potentials. (**D**–**F**) Transepithelial resistance (R_te_), transepithelial voltage (V_te_), and short-circuit current (I_sc_) before and after the addition of ouabain.

**Figure 5 ijms-21-02074-f005:**
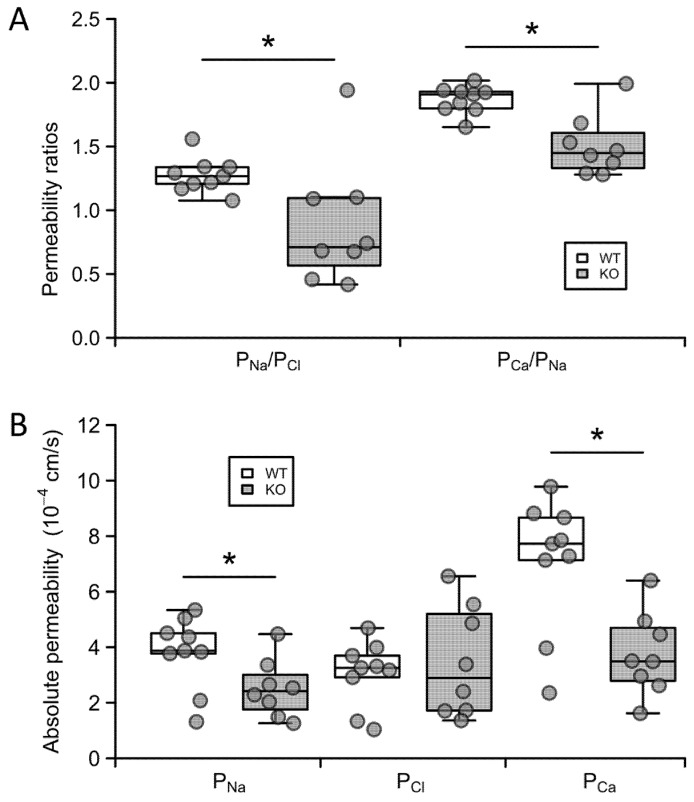
Ion permeabilities in proximal tubules from claudin-12 knockout (KO) and wild type (WT) mice. (**A**) Sodium–chloride and calcium–sodium permeability ratios. (**B**) Absolute permeabilities to sodium, chloride, and calcium. *—*p* < 0.05.

**Figure 6 ijms-21-02074-f006:**
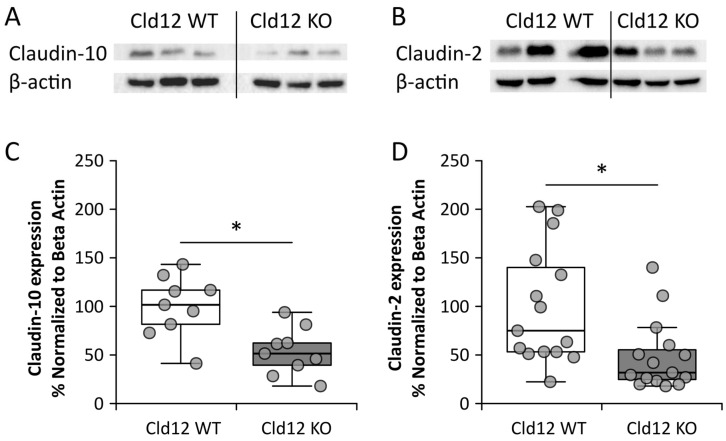
Claudin-12 wild type (WT) and knockout (KO) mouse renal cortical protein expression. (**A**,**C**) Claudin-10 and (**B**,**D**) Claudin-2. * *p* < 0.05.

**Figure 7 ijms-21-02074-f007:**
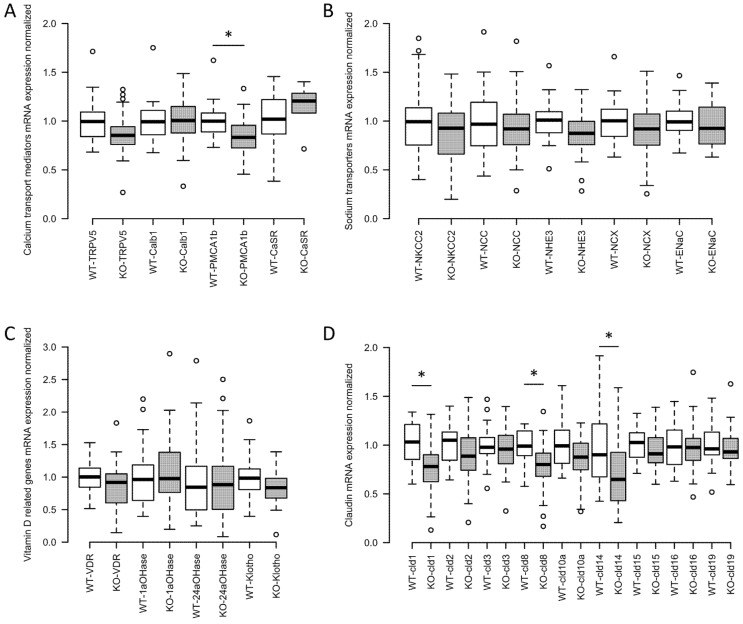
Claudin-12 wild type (WT) and knockout (KO) mouse renal mRNA expression of key genes involved in Ca^2+^ transport. (**A**) Calcium transport mediators: transient receptor potential cation channel subfamily V member 5 (TRPV5), calbindin 1 (Calb1), plasma membrane calcium ATPase 1b (PMCA1b), and calcium sensing receptor (CaSR). (**B**) Sodium transporters: sodium–potassium–chloride cotransporter 2 (NKCC2), sodium–chloride cotransporter (NCC), sodium-hydrogen exchanger 3 (NHE3), sodium–calcium exchanger (NCX), and epithelial sodium channel (ENaC). (**C**) Vitamin D related genes: vitamin D receptor (VDR), 25-hydroxyvitamin D 1 alpha hydroxylase (1aOHase), 24-hydroxylase (24aOHase), and klotho. (**D**) Renal claudins involved in cation transport normalized to 18S as house-keeping gene: claudins 1, 2, 3, 8, 10a, 14, 15, 16, and 19. *n* = 26 and 35, respectively. *—significant with Benjamini–Hochberg critical value for false discovery rate of 0.05. ° represents values that are above or below 1.5 times the interquartile range.

**Figure 8 ijms-21-02074-f008:**
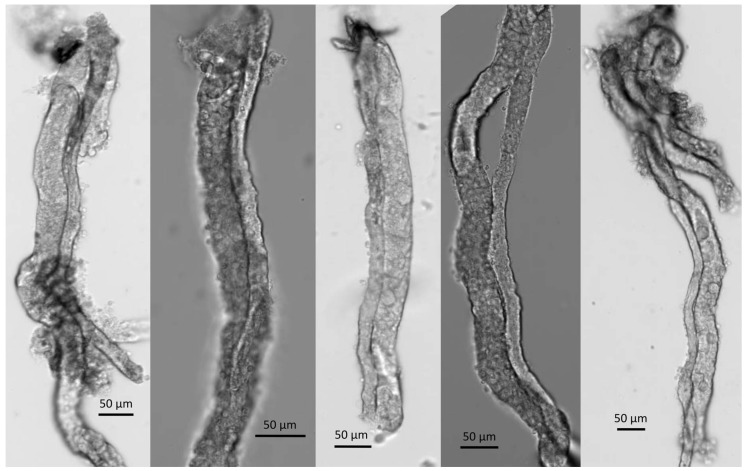
Examples of cTAL (thinner tubule) attached to the straight portion of the pars recta of the proximal tubule (thicker tubule).

**Table 1 ijms-21-02074-t001:** Metabolic cage data.

	WT (*n* = 12–13)	KO (*n* = 17–19)	*p*-Value
Female Weight, g	26.8 ± 1.3 (*n* = 10)	26.6 ± 0.7 (*n* = 9)	0.918
Male Weight, g	29.4 ± 0.9 (*n* = 3)	30.9 ± 0.8 (*n* = 10)	0.337
H_2_O drunk, mL/24 h	6.0 ± 0.4	6.3 ± 0.3	0.426
Chow eaten, g	4.8 ± 0.2	5.3 ± 0.2	0.068
Ca^2+^_ingested_ ^1^, µmol/24 h	1138 ± 46	1266 ± 46	0.068
Urine volume, mL/24 h	1.6 ± 0.2	1.7 ± 0.2	0.603
Fecal excretion ^2^, g/24 h	5.8 ± 0.3	6.4 ± 0.3	0.162
Fecal calcium excretion			
Ca^2+^_feces_, µmol/24 h	1065 ± 53	1116 ± 57	0.536
Ca^2+^_feces_/Ca^2+^_ingested_	0.94 ± 0.04	0.88 ± 0.03	0.177

^1^ Calculated from chow eaten per animal. ^2^ Dry feces weight.

**Table 2 ijms-21-02074-t002:** Serum values.

	WT (*n* = 12–13)	KO (*n* = 17–19)	*p*-Value
Na^+^, mmol/L	150.7 ± 1.2	150.9 ± 0.6	0.820
K^+^, mmol/L	4.7 ± 0.2	4.6 ± 0.2	0.784
Cl^−^, mmol/L ^1^	119 ± 1	117 ± 0.7	0.180
Ca^2+^, mmol/L	2.4 ± 0.2	2.5 ± 0.1	0.369
Cr, µmol/L	0.030 ± 0.004	0.036 ± 0.004	0.315
Glucose, mmol/L	8.8 ± 0.5	9.3 ± 0.4	0.389
BUN, mmol/L ^1^	29.2 ± 1.5	29.3 ± 1.5	0.954
PTH, pg/mL	421 ± 85	332 ± 52	0.370
FGF23, pg/mL	273 ± 16	306 ± 14	0.173
Vitamin D3, pg/mL	81 ± 15	80 ± 11	0.971

^1^ WT *n* = 6, age (WT&KO) between 3 and 6 months.

**Table 3 ijms-21-02074-t003:** Urinary ion excretion.

	WT (*n* = 12)	KO (*n* = 13)	*p*-Value
Na^+^/Creatinine	31 ± 8	39 ± 9	0.495
Cl^−^/Creatinine	140 ± 45	136 ± 33	0.951
K^+^/Creatinine	99 ± 25	93 ± 21	0.855
Ca^2+^/Creatinine	0.24 ± 0.05	0.22 ± 0.04	0.819
PO_4_^3-^/Creatinine	5.4 ± 0.8	5.0 ± 0.7	0.688
Creatinine, µmol/24 h	12 ± 2	16 ± 3	0.377

**Table 4 ijms-21-02074-t004:** Urine ion excretion normalized to the creatinine of WT and KO mice on different calcium-containing diets.

	Normal			High Ca			Low Ca		
	WT (*n* = 12)	KO (*n* = 8)	*p*-Value	WT (*n* = 14)	KO (*n* = 15)	*p*-Value	WT (*n* = 15)	KO (*n* = 11)	*p*-Value
Urine Ca^2 +^/Cr	0.88 ± 0.08	0.79 ± 0.07	0.456	2.65 ± 0.48	3.10 ± 0.52	0.555	0.62 ± 0.03	0.60 ± 0.01	0.614
FE Ca	0.63 ± 0.15	0.37 ± 0.21	0.323	1.88 ± 0.46	1.67 ± 0.31	0.703	0.55 ± 0.14	0.31 ± 0.07	0.178

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
