# Peer review of "Claudin-12 Knockout Mice Demonstrate Reduced Proximal Tubule Calcium Permeability"

_ijms, 2020, doi:10.3390/ijms21062074_

Round 1

Reviewer 1 Report

The revised manuscript submitted by Plein et al. describing the role of claudin-12 in proximal tubule paracellular transport has been vastly improved by additional substantiate experimentation, especially the calcium loading animal experiments and the examination of other claudin isoforms. Now the conclusions made have been strengthened. Despite this, there are still issues that have not been fully addressed or require explanation:

Immunostaining to locate claudin 12 in the nephron segments in Fig. 2 are still very unsatisfactory. Only one glomerulus is seen in the images (which is usually a histological standard to locate proximal tubules), the nephron segment marker stains are not acceptable (in particular the DCT and CD where essentially no staining is seen) and higher magnification panels were not included. Please rectify as this figure is imperative to the manuscript, which claims claudin 12 is expressed exclusively in the proximal tubule.

On p. 5, lines 8-16, the use of ouabain is mentioned. Please indicate the target of ouabain. For the naïve reader, this text can be confusing since paracellular transport is actually dependent on transcellular transport (as described in the Introduction). Thus if transcellular transport is blocked through the ouabain-sensitive Na/K-ATPase, how can paracellular transport be measured? Of course the permeability measurements are based on ion diffusion gradients. A few additional explanatory sentences would make it easier for the non-epithelial specialist to follow.

Though not statistically different, there is a tendency for a decrease in PTH and an increase in FGF23 that could be participating in the observed normalization of serum calcium levels. Please discuss the putative roles of these hormones.

The title should reflect the conclusions of the manuscript.

Author Response

Reviewer #1:

The revised manuscript submitted by Plain et al. describing the role of claudin-12 in proximal tubule paracellular transport has been vastly improved by additional substantiate experimentation, especially the calcium loading animal experiments and the examination of other claudin isoforms. Now the conclusions made have been strengthened. Despite this, there are still issues that have not been fully addressed or require explanation:

Thank you. We agree the manuscript was improved and now is further improved by addressing the concerns listed below.

Immunostaining to locate claudin 12 in the nephron segments in Fig. 2 are still very unsatisfactory. Only one glomerulus is seen in the images (which is usually a histological standard to locate proximal tubules), the nephron segment marker stains are not acceptable (in particular the DCT and CD where essentially no staining is seen) and higher magnification panels were not included. Please rectify as this figure is imperative to the manuscript, which claims claudin 12 is expressed exclusively in the proximal tubule.

We are sorry for the confusion. We are not claiming that claudin-12 is expressed exclusively in the proximal tubule, but that expression is greatest there. Moreover, we are only claiming that mRNA expression is greatest in the proximal tubule due to the lack of a good antibody against claudin-12. We have clarified this in the paper pages 4, lines 18-19 and page 13 lines 1-2. However, we sought to improve the described images, and in keeping with the reviewers suggestion have repeated the co-staining of Xgal, a reporter of claudin-12 mRNA expression with AQP1, CAII and NCC, and supplied digitally zoomed images of less tubules for each – see updated Fig 2. I hope this is convincing that there is very little if any reporter expression in NCC and CAII positive tubules.

On p. 5, lines 8-16, the use of ouabain is mentioned. Please indicate the target of ouabain. For the naïve reader, this text can be confusing since paracellular transport is actually dependent on transcellular transport (as described in the Introduction). Thus if transcellular transport is blocked through the ouabain-sensitive Na/K-ATPase, how can paracellular transport be measured? Of course the permeability measurements are based on ion diffusion gradients. A few additional explanatory sentences would make it easier for the non-epithelial specialist to follow.

 We have added two sentences – page 5 lines 16 -19 to clarify.

Though not statistically different, there is a tendency for a decrease in PTH and an increase in FGF23 that could be participating in the observed normalization of serum calcium levels. Please discuss the putative roles of these hormones.

 We have added this to the discussion page 12, lines 14 – 22.

The title should reflect the conclusions of the manuscript.

We have changed the title to be more precise, it is now “Claudion-12 knockout mice demonstrate reduced proximal tubule calcium permeability”

Reviewer 2 Report

The authors addressed my previous comments.  I have no further issues with the study.

Author Response

The reviewer wrote:

 "The authors addressed my previous comments.  I have no further issues with the study."

Excellent, thank you.

Round 2

Reviewer 1 Report

The authors' additional efforts have improved the manuscript to publication quality. No further comments.

Minor: Please check the legend of Figure 3 for spelling errors.

This manuscript is a resubmission of an earlier submission. The following is a list of the peer review reports and author responses from that submission.

Round 1

Reviewer 1 Report

The manuscript by Plain and colleagues “Claudin-12 contributes calcium permeability to the renal proximal tubule” investigates the functional consequences of Claudin-12 deletion on systemic Ca2+ balance and urinary electrolyte excretion in mice.  Despite the results being mostly negative, this is a well-written and diligently performed study using a variety of experimental techniques, including direct measurements of paracellular permeability in the perfused proximal tubules.  The experiments are properly detailed and the conclusions are reasonable.  The referee can only offer minor suggestions to improve presentation of the provided Figures.  In summary, I would like to complement the authors on performing an excellent study.

Minor suggestions:

Figure 1C: please provide the size values (in bp);

Figure 2 and 3: scale bars are necessary;

Figure 4B: it is unclear whether the provided original trace was from WT or KO animal.  It might be beneficial to include both conditions for better visual appreciation and comparison.

Figure 5: Figure legend should be expanded to contain the description of the tested genes.  Moreover, the meaning of the open circles is not known.

Reviewer 2 Report

The study by Plain et al. investigates the role of proximal tubule expressing tight junction protein claudin-12 in the paracellular reabsorption of calcium and sodium by the kidney proximal tubule and have generated a knockout mouse model for claudin-12. Using immunohistochemical staining of kidney tissue sections, the authors attempt to localize claudin-12 to the proximal tubule. Functional studies using microperfused straight proximal tubules evidence decreased permeability to calcium and sodium in the knockout animals. However, no changes were observed in calcium levels in the urine as well as in the levels of hormones which regulate blood calcium in heterozygotes vs. wildtype mice suggesting that changes in tubular electrolyte permeability through loss of claudin-12 can be compensated. The study suffers from major limitations and methodological flaws. Furthermore, prior evidence that claudin-12 is present in the renal proximal tubule, and may have a functional role there, is based solely on PCR data and only from the current group. Finally, some of the claims made in the manuscript are not fully substantiated by supporting data, e.g. presence of claudin-12 in straight proximal tubule.

It is not explained why the authors continued their functional studies using the heterozygotes rather than the total knockout mice. What was the phenotype of these mice? The age of the mice at the time of experimentation is also not indicated. Did the authors try challenging the mice with a high calcium or low calcium diet?

What is the impact of claudin-12 knockout on the expression of other claudins (preferably at protein level) in the renal proximal tubule? Only total kidney claudin mRNA was analyzed in Figure 5.

Does claudin-12 colocalize with claudin-2, an established renal proximal tubule tight junction protein?

Is transcellular transport affected by claudin-12 knockout?

Figure 2: It is not indicated in the Figure Legend that the higher magnification Figure 2C is derived from Figure 2B. Please indicate area of magnification on 2B, label relevant tubular structures in 2C and add scale bars to images. Is there different expression in the convoluted and straight segments of the proximal tubule? This would be an interesting aspect to investigate since based on the manuscript’s hypothesis, the S1 segment (where the most paracellular resorption takes place) should express higher amounts of claudin-12. A number of proximal tubule S1/S2/S3 segment-specific markers are well established.

Figure 3 images also require scale bars as well as negative controls for the tubule-segment specific markers. Please mention that AQP1 also stains the loop of Henle (thin descending limb) therefore positive signals are seen in the medulla. The stainings for NCC and CAII are not convincing. CAII is an unconventional marker for the collecting duct. Usually AQP2, the proton pump or bicarbonate/chloride exchanger are used. Calbindin could also be used for the distal tubule. Maybe there are better antibodies for these proteins. Higher magnifications for detail would be beneficial.

p.6, line 9: In my opinion, the permeabilities for Na and Ca are altered similarly in the KO animals compared to WT, both are reduced by about 50%. Thus, it is questionable whether it can be concluded that the permeability for Ca is more affected than Na permeability by claudin-12 knockout.

Did the authors examine expression levels of the CaSR?

The data in Table 3 are presented in an incorrect manner. The GFR and fractional excretion should be determined (the authors may want to refer to Gong Y et al. EMBO J 2012 where claudin-14 in the TAL was investigated).

Molecular weight is not indicated in Figure 1C.

p.14, line 24: Please correct to Goldman-Hodgkin-Katz.